# Towards sustainable pharmacy practice: Community pharmacists' experiences with medicine waste, reuse and disposal

**Kingston Rajiah** *, **Ellen McLaughlin**

School of Pharmacy and Pharmaceutical Sciences, Ulster University, Coleraine, United Kingdom

* kingrajiah@gmail.com

## Abstract

### Objective

To explore the perspectives and experiences of community pharmacists in Northern Ireland regarding returned medicines, pharmaceutical waste, and safe disposal practices, with a focus on identifying barriers, opportunities, and policy implications.

### Methods

A qualitative study using semi-structured in-depth interviews was conducted with 15 community pharmacists recruited through purposive sampling. Interviews were held online using Zoom or Microsoft Teams, lasting 30–45 minutes each. Data was audio-recorded, transcribed verbatim, and analysed using Braun and Clarke's six-step inductive thematic analysis framework. Two researchers independently coded the data, and discrepancies were resolved through discussion.

### Results

Seven key themes were identified: (1) Equitable access to return services, highlighting variability in public awareness and participation; (2) Safe pharmacy disposal practices, reflecting structured yet resource-dependent protocols; (3) Waste reduction through supply control, where pharmacists actively worked to minimise unnecessary dispensing; (4) Collaborative waste management, revealing informal partnerships with GPs and care homes; (5) Financial/Operational Burden of Disposal Services, showing frustration at the destruction of potentially reusable medications; (6) Workforce skills and support, noting a lack of formal training and clear guidance; and (7) Legal Tensions in Medicine Reuse, capturing the stiffness between waste reduction and strict safety regulations.

**Data availability statement:** Yes - all data are fully available without restriction; All relevant data are within the manuscript and its Supporting Information files. For further data requests, please contact Aaron Courtenay at a.courtenay@ulster.ac.uk.

**Funding:** The author(s) received no specific funding for this work.

**Competing interests:** The authors have declared that no competing interests exist.

## Conclusion

Community pharmacists in Northern Ireland are committed to reducing medicine waste and ensuring safe disposal, but face regulatory, operational, and workforce-related barriers. Addressing these challenges will require standardised protocols, pharmacist-led public education, and greater integration into prescribing processes. These actions directly support global efforts toward achieving the United Nations Sustainable Development Goals, including Good Health and Well-being, Responsible Consumption and Production, and Climate Action. Empowering pharmacists through training, policy support, and interprofessional collaboration is essential for building a more sustainable, equitable, and environmentally responsible healthcare system.

## Introduction

Pharmaceutical waste, such as unused, expired, or returned medications, is increasingly recognised as a global health policy concern, characterised by polypharmacy and high prescription volumes [1,2]. The World Health Organisation warns that inappropriate disposal methods, including flushing medications or discarding them in household waste, can lead to pharmaceutical pollution, inefficiencies in healthcare systems, and long-term ecological harm [3].

In the United Kingdom (UK), annual medicine wastage costs the National Health Service (NHS) around £300 million annually, much of which is related to surplus stock held by patients, returned prescriptions, and unutilised medications [4]. This excludes the cost of disposal services and environmental mitigation. As healthcare budgets tighten and sustainability becomes a global imperative, reducing pharmaceutical waste is essential to developing efficient and environmentally responsible health systems. Community pharmacies play a pivotal role in mitigating medication waste [5,6]. As the point of contact for patients receiving medicines, pharmacists are uniquely positioned to prevent and reduce waste at several points in the medicines lifecycle, from prescribing and dispensing to patient adherence, returns, and disposal [7]. To illustrate this, Fig 1 shows key stages in the pharmaceutical supply and use process where waste may occur, and where pharmacists can intervene to improve sustainability through medicines optimisation, patient counselling, return schemes, and safe disposal.

Yet, in regions like Northern Ireland, a lack of unified operational guidance has led to inconsistent practices across pharmacies. Existing literature estimates that unused medicines cost the Northern Ireland health system around £18 million each year, with approximately 165 tonnes of medicines wasted annually [8,9].

Medicine wastage arises from multiple sources, including over-prescribing, patient non-adherence, changes to therapy, and mortality, with studies estimating 20% of returned medicines are unopened, highlighting the significant inefficiency in the system [10], but they cannot be reused due to strict safety, storage, and anti-counterfeit regulations [11]. While the possibility of redistribution has been raised, it remains a contested area due to concerns over patient safety, contamination, and regulatory

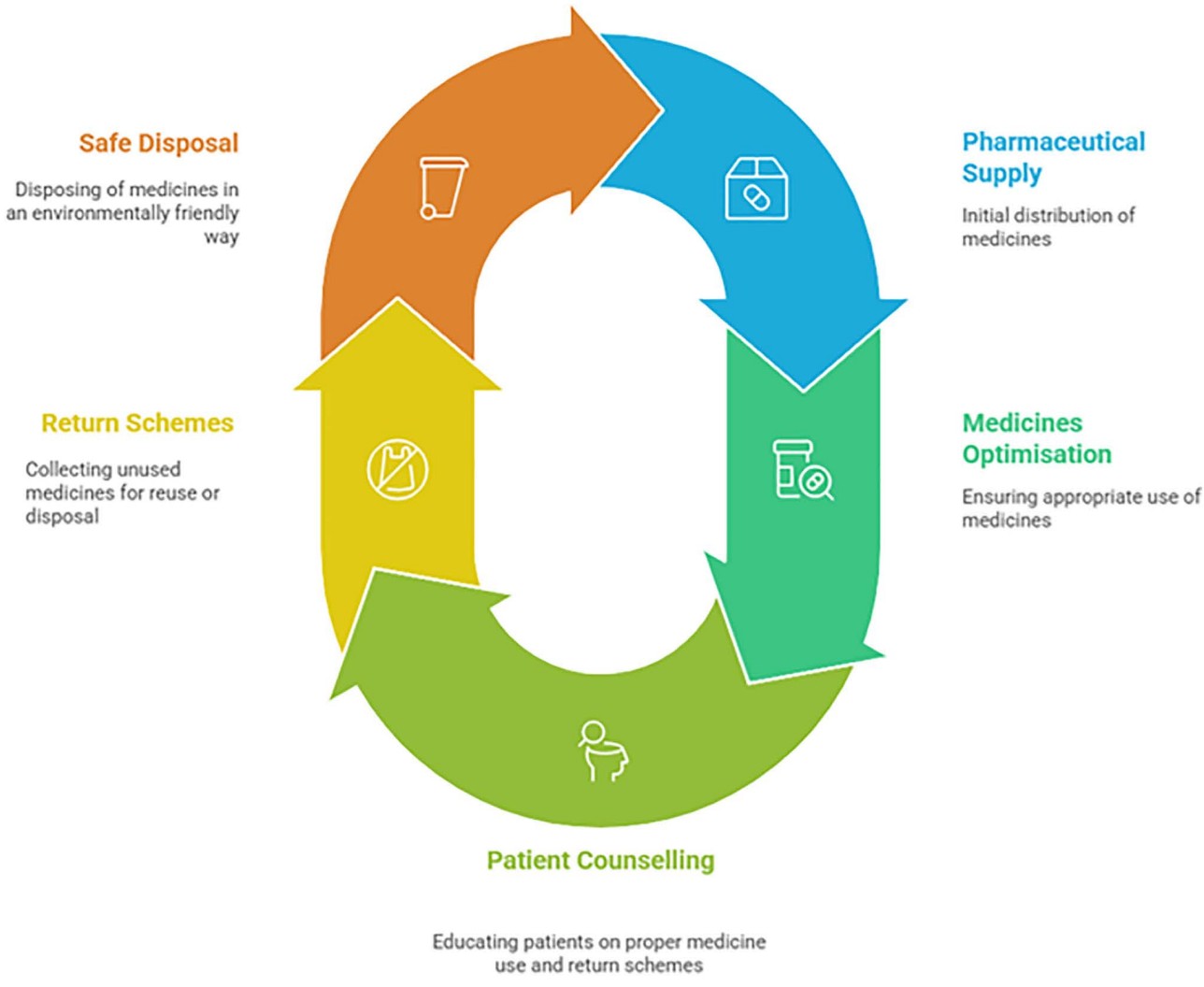

**Fig 1. Medicine lifecycle- environmental sustainability.**

liability. The environmental impact of pharmaceutical waste is increasingly concerning. Active pharmaceutical ingredients have been found in rivers, drinking water, and soil, particularly antibiotics, hormones, and cytotoxic agents [12,13]. These residues threaten aquatic ecosystems and contribute to antimicrobial resistance. Public disposal practices remain inconsistent; research from the UK and EU reports that many people continue to dispose of medicines in sinks or toilets due to a lack of awareness or confusion about proper return schemes [14, 15].

In Northern Ireland, the regulation of medicines is managed by the Department of Health, Medicines and Healthcare products Regulatory Agency (MHRA), and Pharmaceutical Society of Northern Ireland (PSNI). Unlike in England or Scotland, there is no single unified statutory protocol that governs how returned or unused medicines should be managed at the community level. This is due to a combination of distinct legal traditions, devolved governance, and a fundamentally different approach to the organisation of services. Instead, practices are guided by general professional standards and legislation relating to waste handling, such as the Waste Management Licensing Regulations (NI) 2003 and the Controlled Waste Regulations (NI) 2002. These rules establish thresholds for storing pharmaceutical waste and outline legal requirements for classification, transport, and destruction.

Pharmacies in Northern Ireland operate under the Paragraph 39 exemption, allowing them to store up to five cubic metres of returned medicines for up to six months. Disposal is generally handled by licensed waste contractors, often funded by the Department of Health. However, in the absence of detailed local policies or operational guidance, individual pharmacies must interpret these frameworks themselves. This results in variation in practice, shaped by staffing levels, storage capacity, local prescribing behaviours, and pharmacists' professional discretion. The geography of Northern Ireland is marked by rural isolation, transport limitations, and a centralised prescription system, which both complicate and enable waste reduction strategies. In rural areas, pharmacists often develop long-term relationships with patients, which can facilitate medicine optimisation. However, the same rurality can limit access to disposal schemes and create logistical barriers to regular waste collection. Despite these barriers, evidence suggests community pharmacists are willing to expand their role in reducing medicine waste [16]. Low-cost interventions, such as structured counselling, reminder services, and pharmacist-led medication reviews, have proven effective in minimising over-supply and unnecessary dispensing. However, such efforts require clearer policy mandates, adequate staffing, and interprofessional collaboration. Despite the urgency of pharmaceutical waste, including its environmental, economic, and public health dimensions, empirical data specific to Northern Ireland are scarce. Most existing studies extrapolate from broader UK contexts or environmental monitoring reports, potentially overlooking the region's distinct regulatory and cultural landscape.

Comparatively, other high-income countries have implemented different strategies. For example, Australia operates a federally funded Return Unwanted Medicines (RUM) program, while parts of the United States allow for take-back programs in pharmacies and law enforcement sites. European countries like the Netherlands and France have centralised pharmaceutical waste protocols integrated into public awareness campaigns. These models contrast with Northern Ireland's less standardised, pharmacy-led approach and highlight both challenges and opportunities for improvement.

This research took place among community pharmacists working across Northern Ireland, encompassing urban and rural locations. Northern Ireland's healthcare system integrates community pharmacies into public health service delivery, operating within a centralised prescription model and a mix of independent and corporate pharmacies. However, there is no standardised protocol for the handling of returned or wasted medicines, resulting in variability in how these processes are managed at the local level. This variation provided a valuable opportunity to explore different operational approaches, identify common challenges, and consider the potential for more sustainable, system-wide practices in medicine waste management. This study explored community pharmacists' experiences with returned medicines, medicine wastage, and safe disposal in Northern Ireland. It sought to identify operational challenges, assess current practices, and gather recommendations to inform policy, improve pharmacy services, and enhance public engagement on these critical issues.

## Materials and methods

### Ethical approval

This study was performed in accordance with the ethical principles of the Declaration of Helsinki. Ethical approval was obtained from the School of Biomedical Sciences Ethics Filter Committee (FCBMS) at Ulster University (Approval No: FCBMS-24–167-A). All participants completed a written informed consent prior to taking part in the study. Anonymity and confidentiality were maintained throughout.

### Study design

This study adopted a qualitative descriptive design, guided by an inductive thematic analysis approach as outlined by Braun and Clarke [17]. The design was selected to address the research aim of exploring community pharmacists' perspectives and experiences regarding medicine returns, medicine wastage, and safe disposal in Northern Ireland, and to identify context-specific recommendations for improved practice. The qualitative descriptive approach was considered appropriate as it provides a straightforward means of capturing and presenting participants' views in their own words, particularly in areas where little prior research exists. Thematic analysis offered a flexible yet rigorous method for identifying,

analysing, and reporting patterns within the data, allowing the researchers to develop a rich understanding of pharmacists' operational realities without imposing pre-existing theoretical frameworks.

## Participant recruitment

A purposive sampling strategy was used to ensure the inclusion of pharmacists from a range of ages, workplace settings (independent and chain pharmacies), geographical locations (urban, semi-urban, and rural), and professional backgrounds (varying years of experience and both managerial and non-managerial roles). To be eligible, participants had to be registered pharmacists working in community pharmacy in Northern Ireland, have at least one year of post-registration experience, and be directly involved in the handling of returned medicines and disposal processes. Recruitment was carried out through professional pharmacy networks, email invitations distributed via representative bodies, and direct contact with known pharmacy leads. No financial incentives were offered for participation. In total, 15 pharmacists took part in the study. Thematic saturation, defined as the point at which no new codes were identified during analysis, was reached by the thirteenth interview. Two further interviews were conducted to confirm the stability of the themes. Participant demographic information, including age, gender, years of experience, and type of pharmacy, is provided in Table 1 to aid interpretation and transferability.

## Data collection

Data were gathered through semi-structured, in-depth interviews conducted online using Zoom or Microsoft Teams, depending on participant preference. This approach offered flexibility and accessibility for participants based in rural areas. The interview guide was developed following a review of literature on pharmaceutical waste, environmental sustainability, and public health practice. It covered eight broad areas, including current medicine return procedures, staff training and guidance, public engagement, regulatory awareness, environmental concerns, operational barriers, and suggestions for policy improvement. The guide was piloted with two pharmacists, resulting in refinements to improve clarity

**Table 1. Demographics of participants.**

| Demographic Characteristics | Frequency (n = 15) | Proportion % |
|---|---|---|
| Age Range in years | | |
| 18-30 | 4 | 26.67 |
| 31-40 | 6 | 40.00 |
| 41 and above | 5 | 33.33 |
| Gender | | |
| Female | 9 | 60.00 |
| Male | 6 | 40.00 |
| Experience in years | | |
| 1–5 | 4 | 26.67 |
| 6-10 | 3 | 20.00 |
| 11-20 | 6 | 40.00 |
| Above 20 | 2 | 13.33 |
| Job description | | |
| Pharmacist | 9 | 60.00 |
| Pharmacist Manager | 6 | 40.00 |
| Pharmacy location | | |
| Urban | 7 | 46.67 |
| Rural | 8 | 53.33 |

and flow, including the simplification of technical terms and reordering of questions to support a more natural conversation. Field notes were made immediately after each interview to capture contextual details such as tone, emphasis, and non-verbal cues where visible. Interviews lasted between 30 and 45 minutes, were audio-recorded with the participant's informed consent, and were transcribed verbatim. All transcripts were anonymised during transcription to protect participant confidentiality. Interviews were conducted between January and May 2025.

## Data analysis

Thematic analysis was undertaken following Braun and Clarke's six-phase framework, adopting an inductive and semantic approach [17]. This meant that coding was driven by the data itself rather than a pre-existing theory, and that codes reflected the explicit meanings expressed by participants. Analysis began with familiarisation, where all transcripts were read multiple times by one of the researchers (EM), accompanied by reflective note-taking to record early impressions and potential patterns. Initial coding was conducted independently by the researcher (EM), working line-by-line through the transcripts to identify meaningful segments of data. Each segment was assigned a short descriptive label, forming the basis of a shared codebook. This codebook was updated iteratively through weekly meetings, where the research team (EM and KR) discussed, refined, merged, or split codes as necessary. Codes that did not initially fit with emerging patterns were set aside for later review. Once coding was complete, related codes were grouped into themes through an iterative clustering process. Visual mapping techniques were used to explore relationships between codes, and related concepts were brought together into broader categories (refer to Fig 2). For example, codes relating to "lack of formal training" and "No standardised protocols" were eventually combined under the theme "Work skills and support." Themes were reviewed by EM and KR for internal consistency and distinctiveness, with sub-themes identified to highlight the different layers of meaning within the data. We (EM and KR) also looked for examples that did not fit the main patterns and used these to refine themes, broaden their scope, or highlight contrasting viewpoints in the findings. Each theme was then defined and named in a way that captured its core meaning and scope. Saturation was assessed during coding, with no new codes emerging after the thirteenth interview. The remaining two interviews were used to confirm thematic stability. In the write-up, we (EM and KR) combined our interpretation of the data with direct quotes from participants, choosing examples that showed views and opinions. All coding was done manually using Microsoft Word, without the use of qualitative software. Any differences in coding were resolved through consensus, and peer debriefing with the research team (EM and KR) took place every two weeks to challenge interpretations and maintain reflexivity.

## Trustworthiness and rigour

The study adhered to Lincoln and Guba's four criteria for trustworthiness: credibility, transferability, dependability, and confirmability [18]. Credibility was supported through investigator triangulation, regular peer debriefing, and the use of direct participant quotations to ground the analysis. Transferability was facilitated by providing a detailed description of the research setting, participant demographics, and study context. Dependability was ensured by maintaining an audit trail of coding decisions, codebook iterations, and theme refinement. Confirmability was enhanced through reflexive journaling, the use of negative case analysis, and secure retention of anonymised raw data for verification purposes. Member checking was considered but not undertaken, as the ethics committee advised it could compromise participant confidentiality in a small professional community and place additional time burdens on participants already under significant workload pressures. However, the research team engaged in repeated discussions to ensure that the final themes accurately reflected the data.

## Reflexivity

The chief researcher (KR) is a pharmacy professional with experience in public health, pharmacy practice, and qualitative research. Another researcher (EM) brought expertise in environmental health, pharmacy education, and qualitative

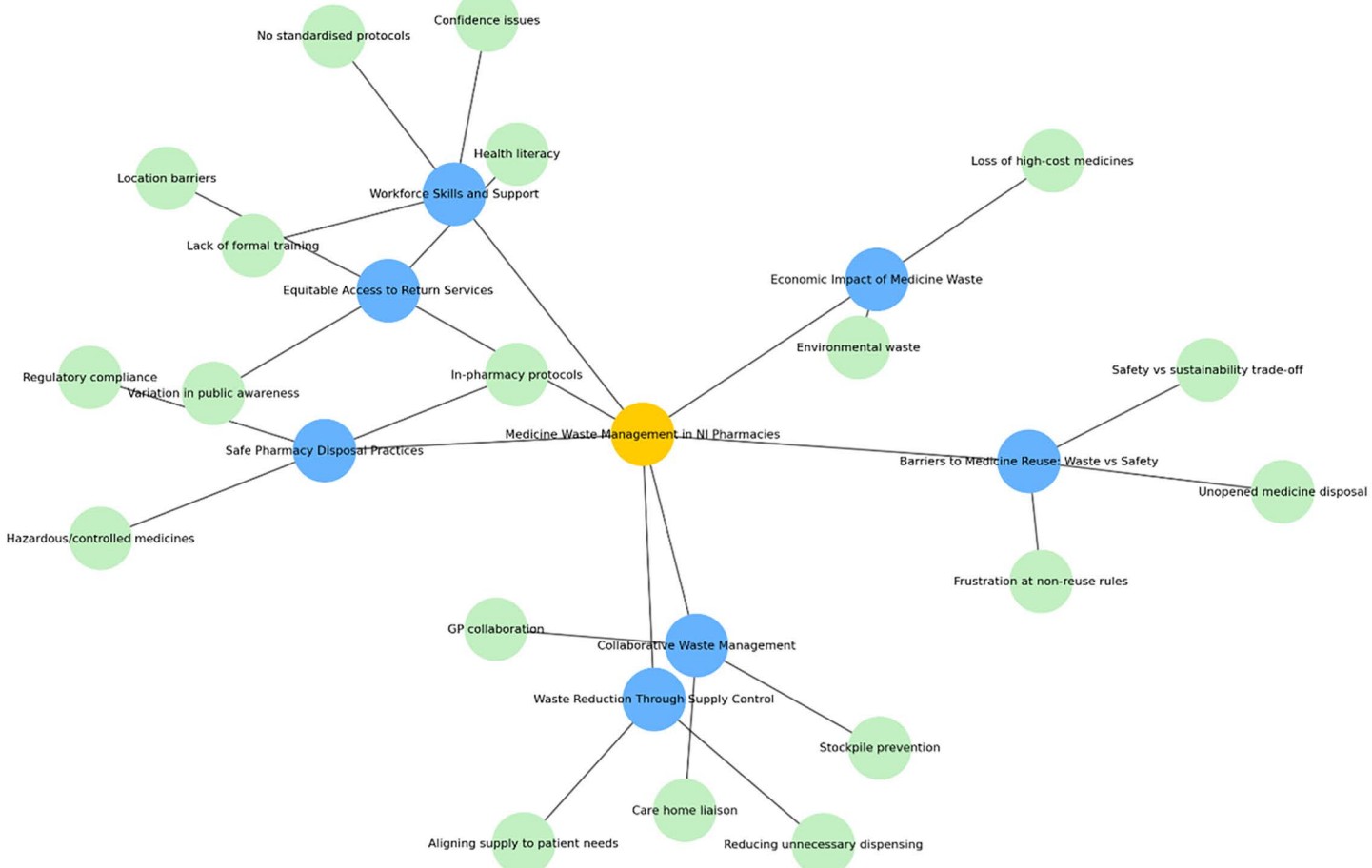

**Fig 2. Visual mapping to explore relationships between codes and related concepts.**

methodology. While these professional backgrounds provided valuable contextual insight, they also carried the risk of bias in interpreting participants' accounts. To address this, the researcher (EM) maintained a reflexive journal to record assumptions, decisions, and reflections after each interview. This diversity of professional perspectives helped challenge individual interpretations and reduce the risk of bias. Team discussions (EM and KR) actively considered how personal and professional experiences might influence the analysis, and a conscious effort was made to ensure that participants' voices were prioritised in the reporting of findings.

This manuscript adheres to the Standards for Reporting Qualitative Research (SRQR) checklist to ensure methodological transparency and rigour. The completed SRQR checklist is included as an S1 Checklist SRQR.

## Results

Thematic analysis of fifteen interviews with community pharmacists in Northern Ireland generated seven themes related to the management of returned medicines, medicine wastage, and safe disposal practices. These themes reflect the complexity of current practices and highlight areas for improvement in policy, communication, and sustainability.

## Participant characteristics

The demographic details of the participants are shown in Table 1. A total of 15 community pharmacists participated in the study. The age of participants ranged from 18 to over 40 years, with the majority aged between 31–40 years (40.00%), followed by those aged 41 and above (33.33%), and 18–30 years (26.67%). The sample comprised more female (60.00%) than male (40.00%) participants. Regarding professional experience, the largest group had 11–20 years of experience (40.00%), followed by 1–5 years (26.67%), 6–10 years (20.00%), and above 20 years (13.33%). In terms of job roles, 60.00% were practising pharmacists and 40.00% held pharmacist manager positions. Slightly more participants were based in rural settings (53.33%) than in urban locations (46.67%). This demographic diversity provided a broad perspective on medicine return and disposal practices across different levels of experience, roles, and geographical contexts. The seven key themes and their definitions are shown in Table 2.

## Theme 1: Equitable access to return services

Participants described a lack of consistency in public knowledge and use of medicine return services. Participants reported that while some patients regularly return unused or expired medicines, others, particularly those from rural areas or with limited health literacy, may remain unaware of appropriate disposal options. This uneven awareness was viewed as a barrier to both environmental safety and effective waste reduction.

> *"Some people still don't realise that you can return the medicines back to the chemist."* (P5)

> *"Posters don't really work; people respond better to a quick chat."* (P14)

Some pharmacists observed increasing public engagement in recent years but agreed that further education is needed to reach vulnerable groups.

> *"There's definitely more awareness now… but there's still a long way to go."* (P9)

While efforts such as brief conversations and in-store reminders were mentioned, participants highlighted that there is no structured role or system for pharmacists to actively raise public literacy around medicine disposal.

**Table 2. Key themes and their definitions.**

| Theme Title | Definition |
| --- | --- |
| Equitable Access to Return Services | Variation in public knowledge and access to medicine return schemes, influenced by location and health literacy. |
| Safe Pharmacy Disposal Practices | Established in-pharmacy protocols for disposal, especially for hazardous and controlled medicines. |
| Waste Reduction Through Supply Control | Pharmacist-led efforts to reduce unnecessary dispensing and align supply with patient needs. |
| Collaborative Waste Management | Informal collaborations with GPs and care homes to address medicine stockpiling and returns. |
| Financial/Operational Burden of Disposal Services | Perceived financial and operational burden in disposing of unused medicines. |
| Workforce Skills and Support | Lack of formal training and standardised protocols for handling medicine waste safely and efficiently. |
| Legal Tensions in Medicine Reuse | Frustration over regulations that prevent the reuse of unopened medicines, despite the potential for safe redistribution. |

## Theme 2: Safe pharmacy disposal practices

Participants described procedures for the disposal of returned medicines, including the use of designated containers, waste segregation, and licensed third-party disposal services. These processes were particularly strict when dealing with controlled drugs, which require denaturing before disposal.

> *"We have dedicated bins… a licensed waste company comes and collects."* (P3)

> *"Controlled drugs need to be denatured… there's a strict process."* (P2)

Despite these procedures, pharmacists expressed concern about the environmental implications of improper public disposal, such as flushing medicines down sinks or toilets.

> *"Improper disposal can contaminate water supplies or end up in landfills."* (P8)

This theme highlights both the professionalism within pharmacy waste handling and the need for stronger public messaging around environmental risks.

## Theme 3: Waste reduction through supply control

Participants reported that medicine waste often originates from overprescribing, automatic repeat dispensing, or patient non-adherence. Many described active strategies to mitigate such waste, including engaging patients in medication reviews and collaborating with prescribers to minimise unnecessary supply.

> *"We check if patients really need everything before dispensing."* (P6)

> *"We try to work with GPs to reduce overprescribing."* (P4)

Participants expressed frustration that once medicines leave the pharmacy, even if unopened, they cannot legally be reused, which adds to the perception of wastefulness.

> *"Once medicines leave the pharmacy, we can't reuse them… it feels wasteful."* (P2)

This theme reflects pharmacists' desire to contribute more directly to sustainable prescribing and dispensing practices, while also acknowledging regulatory limitations.

## Theme 4: Collaborative waste management

Participants spoke about informal collaborations with GPs, care homes, and local charities to manage medicine waste more efficiently. Some took proactive steps to highlight recurring medicine returns or to support patients who were at risk of medication accumulation.

> *"We sometimes flag things to GPs if a patient is constantly returning meds."* (P7)

Others described wider community engagement efforts but noted the lack of structured systems or shared accountability.

> *"We work with local GP practices, nursing homes, and nonprofit organisations."* (P8)

> *"There isn't much structured collaboration on this issue."* (P5)

This theme highlights opportunities for more formalised, multi-agency partnerships aimed at reducing medicine wastage and supporting safe disposal.

### Theme 5: Financial/operational burden of disposal services

Participants emphasised that medicine disposal is financially and operationally burdensome for pharmacies. Disposal bins, secure storage, and licensed contractor collections incur costs that must be absorbed by the business. In addition to these direct costs, participants described the significant time and staff resources required to sort, log, and prepare medicines for disposal.

*"Safe disposal isn't cheap… we have to factor it into our budget."* (P11)

*"It's time-consuming… someone has to check everything and fill out forms*." (P1)

This theme reflects the economic and operational impact of pharmaceutical waste at the pharmacy level.

### Theme 6: Workforce skills and support

Finally, participants highlighted a need for clearer guidance and greater support in managing medicine waste. Many found existing protocols confusing or inconsistent, particularly when dealing with controlled substances or interpreting new policies.

*"The guidelines can be confusing… I have to double-check them."* (P12)

*"There's a lot of paperwork… and the rules keep changing."* (P7)

The reliance on informal knowledge and experience, rather than formal training, was noted as a vulnerability in maintaining safe and sustainable disposal practices.

*"We mostly rely on experience and common sense."* (P5)

This theme suggests that targeted workforce development, including training and standardised protocols, is essential to ensure safe and efficient medicine return systems.

### Theme 7: Legal tensions in medicine reuse

This theme captures pharmacists' recurring frustration with the legal restrictions that prohibit the reuse of returned, unopened, and unexpired medicines. Participants expressed a strong ethical concern over the destruction of perfectly viable medication, especially high-cost or short-supply items. While acknowledging the importance of patient safety, many pharmacists questioned whether current rules sufficiently balance risk with the growing economic and environmental cost of medicine waste.

*"Once medicines leave the pharmacy, we can't reuse them… it feels wasteful."* (P2)

*"Unopened packs are destroyed… it's a waste of resources."* (P10)

*"It would be great to safely redistribute returned unopened medicines."* (P6)

These views reflect an apprehension between regulatory precaution and professional judgement.

## Discussion

This study explored community pharmacists' experiences and perspectives on medicine returns, pharmaceutical waste, and safe disposal in Northern Ireland. Through thematic analysis of fifteen in-depth interviews, seven key themes were identified: equitable access to return services, safe pharmacy disposal practices, waste reduction through supply control, collaborative waste management, the financial/operational burden of disposal services, workforce skills and support, and legal tensions in medicine reuse. These findings provide valuable insights into the operational, regulatory, and behavioural dimensions of medicine waste management in the community pharmacy context.

The demographic characteristics of the participants reflect a balanced and diverse representation of community pharmacists across Northern Ireland. The range of professional experience, from early career practitioners (1–5 years) to those with over two decades in practice, enabled an understanding of how medicine return and waste management practices have evolved. Notably, the majority of participants were female and between the ages of 31 and 40, aligning with broader workforce trends in community pharmacy, where women now represent a significant proportion of the profession [19]. The inclusion of both pharmacists and pharmacist managers added further depth to the data, with managers often offering insights into operational protocols and policy adherence. Importantly, the near-even split between urban and rural participants provided contrasting perspectives on access to disposal services and community engagement. This diversity strengthened the credibility of the findings and enhanced the transferability of the results to other UK regions with similar pharmacy structures.

Study findings demonstrate that pharmacists operate at several critical touchpoints of the medicine lifecycle, with potential to influence environmental and health outcomes. At the dispensing stage, pharmacists engage with patients to prevent unnecessary supply, aligning with evidence showing that supply control and deprescribing can significantly reduce medicine waste [20,21]. At the use stage, pharmacists act as accessible educators, emphasising the impact of interpersonal communication over printed materials in raising awareness of proper medicine disposal. This reflects findings from other UK and international studies showing that pharmacy-led patient engagement improves medicine return behaviours and reduces unsafe disposal [22–27]. At the disposal stage, pharmacists see themselves as both health professionals and environmental stewards, ensuring medicines are stored, segregated, and disposed of according to regulatory standards. This reflects international literature positioning pharmacies as critical control points in mitigating pharmaceutical contamination in waterways [28–31]. However, pharmacists in this study reported persistent gaps in public awareness and inconsistent access to disposal services, particularly in rural areas, echoing findings from environmental health studies across Europe and North America [14,15,32]. Framing these findings along the medicine lifecycle highlights how pharmacists influence not just the end of the chain but also upstream decisions around dispensing and patient engagement. This aligns with recent calls to position pharmacists as green champions in healthcare systems, with responsibilities that extend beyond compliance to proactive environmental stewardship [22,33,34].

Pharmacists described engaging in informal collaborations with prescribers, care homes, and community organisations to address medicine waste, but these efforts were ad hoc, fragmented, and largely reactive. This reflects other studies showing that lack of structured collaboration remains a major barrier to effective waste reduction strategies [35–37]. Integrated care models and shared digital infrastructure could support more proactive interventions, such as identifying patients at risk of stockpiling or synchronising prescription cycles between GP practices and pharmacies. International evidence shows that multi-agency interventions are more successful than isolated efforts [36]. The findings highlight the systemic nature of pharmaceutical waste, which requires coordination across prescribers, pharmacists, patients, regulators, and waste management services, rather than placing responsibility solely on community pharmacy.

A key insight from this study is the tension between pharmacists' sense of responsibility for reducing waste and the structural constraints they face. The operational and financial burdens associated with medicine disposal are often invisible in policy discussions. This resonates with NHS data estimating over £300 million annual wastage in England [4] and literature highlighting the environmental cost of pharmaceutical incineration [38]. Moreover, pharmacists expressed legal frustration over destroying unopened,

viable medicines. While acknowledging safety concerns, many questioned whether current zero-reuse policies reflect a balanced approach. This echoes a growing body of literature advocating for carefully regulated medicine reuse schemes under controlled conditions [21,39–43]. Compounding these economic and regulatory tensions is a workforce gap. Pharmacists described relying on experiential knowledge to manage waste rather than receiving structured training, consistent with prior findings that sustainability is underrepresented in pharmacy education and Continuous Professional Development (CPD) [33,44–46]. These factors such as cost, regulation, and workforce capacity, interact to shape pharmacists' ability to act as effective agents of change.

The findings of this study point toward the need for system-wide, standardised, and sustainable policy frameworks for medicine return and disposal. A lifecycle approach to pharmaceutical stewardship would position pharmacists as key players at each stage: deprescribing and supply control, patient engagement, safe disposal, and policy advocacy.

## Strengths of the study

A key strength of this study lies in its focus on community pharmacists in Northern Ireland, a region with a distinctive healthcare system and limited prior research on pharmaceutical waste practices. By adopting a qualitative approach, the study captured the real-world perspectives of pharmacists directly involved in medicine returns and disposal. The use of semi-structured, in-depth interviews allowed participants to reflect freely on their experiences, challenges, and ideas for improvement, generating rich and contextually grounded data. Furthermore, the study used Braun and Clarke's thematic analysis framework to ensure a systematic and rigorous interpretation of the data, with independent coding and collaborative theme development enhancing the credibility of findings.

## Limitations of the study

Despite these strengths, several limitations should be acknowledged. First, the sample size, while adequate for thematic saturation, may not fully capture the diversity of pharmacy practice across all regions or organisational types. Second, participation was voluntary and may have attracted pharmacists with stronger opinions or interest in sustainability, introducing potential self-selection bias. Third, while online interviews via Zoom or Microsoft Teams enabled participation across geographical areas, they may have limited observation of non-verbal cues and introduced technical constraints. Additionally, the study did not include perspectives from other stakeholders (e.g., GPs, waste contractors, or regulators), which could have enriched the data and provided a more holistic view of the medicine waste ecosystem.

## Implications for policy and practice

This study has several important implications for healthcare policy and pharmacy practice. First, the findings reinforce the need for nationally standardised protocols and clear guidance for managing returned and expired medicines in community pharmacies. Regulatory bodies such as the PSNI and Department of Health should collaborate with frontline pharmacists to co-develop practical, enforceable SOPs tailored to pharmacy settings. Second, efforts to reduce medicine waste must be coupled with public education campaigns that are community-specific and pharmacist-led, emphasising not just safety but environmental responsibility. Pharmacies should be supported to fulfil their role as educators through appropriate training and resourcing. Third, there is a strong case for integrating pharmacists more formally into repeat prescribing and medication optimisation pathways, allowing them to intervene earlier in the medicines use cycle. This could reduce unnecessary dispensing and support safer disposal practices. Lastly, the study highlights the importance of investing in pharmacy workforce development. Sustainability in medicine use should be embedded into CPD, undergraduate training, and operational planning across pharmacy sectors.

## Recommendations for future research

Further research is needed to evaluate the effectiveness of medicine reuse models under controlled conditions, especially in light of resource constraints and environmental goals. There is also a need to explore multi-stakeholder perspectives, including general practitioners, patients, care home staff, and policymakers, to develop a coordinated waste management

strategy. Quantitative studies estimating the carbon and financial footprint of pharmaceutical waste in community settings would provide valuable data for cost-benefit analyses. Lastly, implementation research evaluating the rollout of pharmacist-led waste reduction interventions, such as enhanced medication reviews or return counselling, could inform future practice and policy.

## Conclusion

This study provides in-depth insights into the real-world challenges and practices surrounding medicine returns, wastage, and safe disposal in Northern Ireland's community pharmacies. Pharmacists are actively engaged in reducing avoidable waste and promoting safe disposal, despite working within fragmented regulatory frameworks and with limited training or structured support. Their frontline role in medicines optimisation positions them as key contributors to a more sustainable and responsible pharmaceutical system. They also voiced frustration at being unable to reuse returned but unopened medicines, highlighting the growing tension between safety regulation and sustainability. These findings align closely with global priorities set out in the United Nations Sustainable Development Goals, particularly Good Health and Well-being, Responsible Consumption and Production, and Climate Action. Advancing sustainability in pharmacy practice will require coordinated efforts in policy, public education, and workforce development, ensuring that pharmacists are empowered to lead medicine waste reduction efforts as part of a broader transition to greener and safer healthcare systems.

## Supporting information

**S1 Checklist. SRQR. Standards for reporting qualitative research.**
(PDF)

**S1 File. Dataset. Minimal anonymised data set.**
(DOCX)

## Acknowledgments

The authors thank Ulster University, the Centre for International Health Innovation & Partnerships, the Social & Administrative Pharmacy Practice Cluster, and the School of Pharmacy and Pharmaceutical Sciences for providing the opportunity and support for this work.

## Author contributions

**Conceptualization:** Kingston Rajiah.

**Data curation:** Kingston Rajiah, Ellen McLaughlin.

**Formal analysis:** Kingston Rajiah, Ellen McLaughlin.

**Investigation:** Ellen McLaughlin.

**Methodology:** Kingston Rajiah, Ellen McLaughlin.

**Project administration:** Ellen McLaughlin.

**Resources:** Kingston Rajiah, Ellen McLaughlin.

**Supervision:** Kingston Rajiah.

**Validation:** Kingston Rajiah, Ellen McLaughlin.

**Visualization:** Kingston Rajiah, Ellen McLaughlin.

**Writing – original draft:** Ellen McLaughlin.

**Writing – review & editing:** Kingston Rajiah.

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
