## [Decision Letter · Decision Letter 0]

2 Oct 2025

PONE-D-25-44528Towards Sustainable Pharmacy Practice: Community Pharmacists’ Experiences with Medicine Waste and DisposalPLOS ONE

Dear Dr. Rajiah,

Thank you for submitting your manuscript to PLOS ONE. After careful consideration, we feel that it has merit but does not fully meet PLOS ONE’s publication criteria as it currently stands. Therefore, we invite you to submit a revised version of the manuscript that addresses the points raised during the review process.

Strengthen introduction with rationale, context, and conceptual figure.Clarify coding process, reflexivity, manual approach, and add COREQ checklist.Distinguish overlapping themes in Results.Reframe discussion with higher-level synthesis and integration with literature.Embed global policy/SDG connections earlier in the paper.Revise conclusion for conciseness and alignment.

We look forward to receiving your revised manuscript.

Kind regards,

Muhammad Shahzad Aslam, Ph.D.,M.Phil., Pharm-D

Academic Editor

PLOS ONE

Journal Requirements:

Additional Editor Comments:

1-Strengthen the rationale and narrative arc. Clarify why this study was conducted, and situate it clearly within global and Northern Ireland contexts.

2-Expand background on pharmacy services in Northern Ireland and how waste/disposal practices compare with other regions.

3-Consider including a figure that illustrates the lifecycle of medicines and potential pharmacist intervention points (deprescribing, disposal, patient education).

4-Line 70: provide more details on the framework mentioned.

5-Explicitly state who performed the coding (using initials) and include this in the reflexivity statement.

6-Clarify whether coding was done manually (e.g., Microsoft Word, NVivo, pen-and-paper).

7-Include the COREQ checklist as a supplementary file and confirm completeness.

8-Distinguish more clearly between Theme 5 (Economic impact of waste) and Theme 7 (Barriers to reuse). Currently, the same quotations are repeated for both, which blurs thematic boundaries. Consider reassigning or expanding quotes to avoid duplication.

9-The discussion needs a stronger synthesis. Currently, it reiterates theme-level findings rather than elevating them into higher-level insights.

10-Integrate findings with wider literature (beyond Northern Ireland) and highlight pharmacists’ potential as environmental stewards—a concept well introduced in lines 352–358 but needing greater emphasis throughout.

11-Link discussion points more explicitly to global policies and UN Sustainable Development Goals.

12- Provide more indepth study limitation to address all potential bias.

13- Expand future direction and provide more details.

Reviewers' comments:

Reviewer's Responses to Questions

**Comments to the Author**

1. Is the manuscript technically sound, and do the data support the conclusions?

Reviewer #1: Yes

Reviewer #2: Yes

2. Has the statistical analysis been performed appropriately and rigorously? 

Reviewer #1: Yes

Reviewer #2: N/A

3. Have the authors made all data underlying the findings in their manuscript fully available?

Reviewer #1: Yes

Reviewer #2: No

4. Is the manuscript presented in an intelligible fashion and written in standard English?

Reviewer #1: Yes

Reviewer #2: Yes

5. Review Comments to the Author

Reviewer #1: This is a well-written and engaging paper on a relevant topic. The use of purposive sampling and manual thematic analysis is appropriate, though I suggest adding a little more detail on how the coding and theme development were carried out to show rigor and transparency. Overall, the study is strong, and I support acceptance with only minor revisions.

Reviewer #2: Thank you for the opportunity to review this manuscript titled “Towards Sustainable Pharmacy Practice: Community Pharmacists’ Experiences with Medicine Waste and Disposal”. I believe the topic is of importance and warrants discussion in the literature. I enjoyed reading the manuscript and have some questions/comments below.

Methods

I found the methods to be robust and very well written.

Line 107-113 - move to introduction as part of your rationale and justification for your study. This is not methods

Make clear who did the coding by using the initials and add this to the reflexivity

Line 166 - coding was done manually… using microsoft word, pen and paper, etc.?

Add Consolidated criteria for reporting qualitative research (COREQ) checklist and confirm it’s complete

Add ethics information

Introduction

The introduction needs to be strengthened and really make it clear what the story is that you are telling and the rationale for your study.

The intro relies on some assumed knowledge that I am unclear the readers will have.

The intro would benefit from a figure and discussion of the cycle of medicines and where pharmacists can be involved to reduce waste - deprescribing, disposal etc.

We need a better understanding of pharmacy in Northern Ireland, then how medication waste practices occur in other regions around the world and the direct comparison to how it is managed in Northern Ireland.

Line 62-63 - the part of the sentence about 'theoretically suitable for redistribution' doesn't align with the rest of the paragraph and introduction, either expand this point or remove it.

Line 70 - what is the framework? provide more details

Line 79-80 - how does the mix of urban and rural and centralized prescription system offer solutions? Also, what are the solutions?

Line 81 - what barriers are you referring to? I am missing the link to this sentence from the previous ones.

Results

Line 220 - Did the pharmacists do anything to raise literacy or awareness?

Theme 5 and 7 Lines 269-272 and 290-295. The same quotes have been used to illustrate the same point (pharmacists find disposal wasteful and want to reuse unopened meds) but under 2 different themes. Theme 5 seems to have 2 main points - (1) cost to the pharmacy to dispose of the returned meds and (2) perspective of wastefulness for destroying unopened boxes. It is unclear how this second point in Theme 5 differs from Theme 7. I encourage the authors to review and make the distinction clearer.

Discussion

Similarly to the introduction, the discussion needs to be strengthened and really make it clear what the story is that you are telling and the positioning within the knowledge to address the reader’s - SO WHAT?

It is of my opinion that the discussion rephrases and focuses on the paper’s results at the theme level too much. I was looking for more of a higher level synthesis of the results and the comparison of that with existing literature. Your manuscript has me thinking of the lifecycle of medicines and all the touchpoints and ways pharmacists can intervene as environmental stewards - deprescribing, reducing stockpiles, disposing correctly etc. This could be the structure and outline for the discussion and how to meaningfully weave your results with existing literature, leading to an implication for practice of standardized policies for the return of meds.

Line 352-358 - this discussion about pharmacists as environmental stewards is brilliant and I was hoping for more of this throughout the discussion.

Conclusion

Line 571-573 - this is the first mention of these global policies, they should be weaved into the introduction and discussion. They are a great justification and rationale as pharmacists as environment stewards.

6. PLOS authors have the option to publish the peer review history of their article (what does this mean?). If published, this will include your full peer review and any attached files.

Reviewer #1: **Yes:**Farwa Tahir

Reviewer #2: **Yes:**Kaitlyn Watson

---

## [Author Response · Author response to Decision Letter 1]

10 Feb 2026

Dear Editor and reviewers,

Thank you for taking the time to review our manuscript. Your thoughtful comments have greatly helped to improve and strengthen the quality of our work. We have addressed all the points you raised, responding to each comment individually. All revisions have been highlighted in red font in the manuscript for your easy review.

Journal Requirements:

Author’s response: We have ensured that our manuscript meets PLOS ONE's style requirements, including those for file naming.

Author’s response: The data presented in the study are stored securely at the School of Pharmacy and Pharmaceutical Sciences, Ulster University. Investigators act as custodians for the data processed and generated by the study, and they are also responsible for access to any information included. The data is available upon request from the corresponding author.

Author’s response: The minimal anonymised data has been uploaded as a Supporting Information file named S2_Dataset

Author’s response: Thank you.

Author’s response: We have included the ethical statement in the ‘Methods’ section of our manuscript.

“Ethical approval

This study was performed in accordance with the ethical principles of the Declaration of Helsinki. Ethical approval was obtained from the School of Biomedical Sciences Ethics Filter Committee (FCBMS) at Ulster University (Approval No: FCBMS-24-167-A). All participants completed a written informed consent prior to taking part in the study. Anonymity and confidentiality were maintained throughout.”

Author’s response: We have included the title for our Supporting Information files at the end of our manuscript, and updated in-text citation to match accordingly.

“S1_Checklist SRQR. Standards for reporting qualitative research

S2_Dataset. Minimal anonymised data set”

Author’s response: Reviewers have no comments to cite specific previously published works.

Additional Editor Comments:

1-Strengthen the rationale and narrative arc. Clarify why this study was conducted, and situate it clearly within global and Northern Ireland contexts.

Author’s response: We have strengthened the rationale and narrative. We have clarified why this study was conducted and explained it clearly within the global and Northern Ireland contexts.

“Comparatively, other high-income countries have implemented different strategies. For example, Australia operates a federally funded Return Unwanted Medicines (RUM) program, while parts of the United States allow for take-back programs in pharmacies and law enforcement sites. European countries like the Netherlands and France have centralised pharmaceutical waste protocols integrated into public awareness campaigns. These models contrast with Northern Ireland’s less standardised, pharmacy-led approach and highlight both challenges and opportunities for improvement.

This research took place among community pharmacists working across Northern Ireland, encompassing urban and rural locations. Northern Ireland’s healthcare system integrates community pharmacies into public health service delivery, operating within a centralised prescription model and a mix of independent and corporate pharmacies. However, there is no standardised protocol for the handling of returned or wasted medicines, resulting in variability in how these processes are managed at the local level. This variation provided a valuable opportunity to explore different operational approaches, identify common challenges, and consider the potential for more sustainable, system-wide practices in medicine waste management.”

2-Expand background on pharmacy services in Northern Ireland and how waste/disposal practices compare with other regions.

Author’s response: We have expanded background on pharmacy services in Northern Ireland and how waste/disposal practices compare with other regions.

“In Northern Ireland, the regulation of medicines is managed by the Department of Health, Medicines and Healthcare products Regulatory Agency (MHRA), and Pharmaceutical Society of Northern Ireland (PSNI). Unlike in England or Scotland, there is no single unified statutory protocol that governs how returned or unused medicines should be managed at the community level. This is due to a combination of distinct legal traditions, devolved governance, and a fundamentally different approach to the organisation of services. Instead, practices are guided by general professional standards and legislation relating to waste handling, such as the Waste Management Licensing Regulations (NI) 2003 and the Controlled Waste Regulations (NI) 2002. These rules establish thresholds for storing pharmaceutical waste and outline legal requirements for classification, transport, and destruction. Pharmacies in Northern Ireland operate under the Paragraph 39 exemption, allowing them to store up to five cubic metres of returned medicines for up to six months. Disposal is generally handled by licensed waste contractors, often funded by the Department of Health. However, in the absence of detailed local policies or operational guidance, individual pharmacies must interpret these frameworks themselves. This results in variation in practice, shaped by staffing levels, storage capacity, local prescribing behaviours, and pharmacists’ professional discretion. The geography of Northern Ireland is marked by rural isolation, transport limitations, and a centralised prescription system, which both complicate and enable waste reduction strategies. In rural areas, pharmacists often develop long-term relationships with patients, which can facilitate medicine optimisation. However, the same rurality can limit access to disposal schemes and create logistical barriers to regular waste collection.”

3-Consider including a figure that illustrates the lifecycle of medicines and potential pharmacist intervention points (deprescribing, disposal, patient education).

Author’s response: We have included a figure that illustrates the lifecycle of medicines and potential pharmacist intervention.

4-Line 70: provide more details on the framework mentioned.

Author’s response: We have revised the sentence as it is not a framework, but medicine is managed by MHRA, PSNI and DoH.

“In Northern Ireland, the regulation of medicines is managed by the Department of Health, Medicines and Healthcare products Regulatory Agency (MHRA), and Pharmaceutical Society of Northern Ireland (PSNI).”

5-Explicitly state who performed the coding (using initials) and include this in the reflexivity statement.

Author’s response: We have explicitly stated who performed the coding (using initials) and include this in the reflexivity statement.

6-Clarify whether coding was done manually (e.g., Microsoft Word, NVivo, pen-and-paper).

Author’s response: We have revised this line as “All coding was done manually using Microsoft Word.”

7-Include the COREQ checklist as a supplementary file and confirm completeness.

Author’s response: As per the journal guidelines, qualitative research studies should be reported in accordance with the Consolidated criteria for reporting qualitative research (COREQ) checklist or Standards for reporting qualitative research (SRQR) checklist. This study reported in accordance with the SRQR checklist. We have confirmed that it is complete and added it as a supplementary file.

8-Distinguish more clearly between Theme 5 (Economic impact of waste) and Theme 7 (Barriers to reuse). Currently, the same quotations are repeated for both, which blurs thematic boundaries. Consider reassigning or expanding quotes to avoid duplication.

Author’s response: Thank you for pointing out this issue. We have revised themes 5 and 7 now.

Theme 5: Financial/Operational Burden of Disposal Services

Theme 7: Legal Tensions in Medicine Reuse

9-The discussion needs a stronger synthesis. Currently, it reiterates theme-level findings rather than elevating them into higher-level insights.

Author’s response: We have revised the entire discussion section.

10-Integrate findings with wider literature (beyond Northern Ireland) and highlight pharmacists’ potential as environmental stewards—a concept well introduced in lines 352–358 but needing greater emphasis throughout.

Author’s response: Thank you for your comment highlighting the need to strengthen the discussion on pharmacists as environmental stewards. We have revised the entire discussion section to place a stronger emphasis on this perspective.

11-Link discussion points more explicitly to global policies and UN Sustainable Development Goals.

Author’s response: We have revised the entire discussion section and stated the UN SDG in the conclusion.

12- Provide more indepth study limitation to address all potential bias.

Author’s response: We have revised the study limitations.

“Limitations of the Study

Despite these strengths, several limitations should be acknowledged. First, the sample size, while adequate for thematic saturation, may not fully capture the diversity of pharmacy practice across all regions or organisational types. Second, participation was voluntary and may have attracted pharmacists with stronger opinions or interest in sustainability, introducing potential self-selection bias. Third, while online interviews via Zoom or Microsoft Teams enabled participation across geographical areas, they may have limited observation of non-verbal cues and introduced technical constraints. Additionally, the study did not include perspectives from other stakeholders (e.g., GPs, waste contractors, or regulators), which could have enriched the data and provided a more holistic view of the medicine waste ecosystem”.

13- Expand future direction and provide more details.

Author’s response: We have expanded the future direction.

“Recommendations for Future Research

Further research is needed to evaluate the effectiveness of medicine reuse models under controlled conditions, especially in light of resource constraints and environmental goals. There is also a need to explore multi-stakeholder perspectives, including general practitioners, patients, care home staff, and policymakers, to develop a coordinated waste management strategy. Quantitative studies estimating the carbon and financial footprint of pharmaceutical waste in community settings would provide valuable data for cost-benefit analyses. Lastly, implementation research evaluating the rollout of pharmacist-led waste reduction interventions, such as enhanced medication reviews or return counselling, could inform future practice and policy.”

Reviewer #1: This is a well-written and engaging paper on a relevant topic. The use of purposive sampling and manual thematic analysis is appropriate, though I suggest adding a little more detail on how the coding and theme development were carried out to show rigor and transparency. Overall, the study is strong, and I support acceptance with only minor revisions.

Author’s response: We have added a little more detail on how the coding and theme development were carried out to show rigour and transparency.

“Analysis began with familiarisation, where all transcripts were read multiple times by one of the researchers (EM), accompanied by reflective note-taking to record early impressions and potential patterns. Initial coding was conducted independently by the researcher (EM), working line-by-line through the transcripts to identify meaningful segments of data. Each segment was assigned a short descriptive label, forming the basis of a shared codebook. This codebook was updated iteratively through weekly meetings, where the research team (EM and KR) discussed, refined, merged, or split codes as necessary. Codes that did not initially fit with emerging patterns were set aside for later review. Once coding was complete, related codes were grouped into themes through an iterative clustering process.”

Reviewer #2: Thank you for the opportunity to review this manuscript titled “Towards Sustainable Pharmacy Practice: Community Pharmacists’ Experiences with Medicine Waste and Disposal”. I believe the topic is of importance and warrants discussion in the literature. I enjoyed reading the manuscript and have some questions/comments below.

Methods

I found the methods to be robust and very well written.

Line 107-113 - move to introduction as part of your rationale and justification for your study. This is not methods.

Author’s response: We have moved those lines to the introduction section.

Make clear who did the coding by using the initials and add this to the reflexivity

Author’s response: We have made clear who did the coding by using the initials and added this to the reflexivity.

Line 166 - coding was done manually… using microsoft word, pen and paper, etc.?

Author’s response: We have revised this line as “All coding was done manually using Microsoft Word.”

Add Consolidated criteria for reporting qualitative research (COREQ) checklist and confirm it’s complete.

Author’s response: As per the journal guidelines, qualitative research studies should be reported in accordance with the Consolidated criteria for reporting qualitative research (COREQ) checklist or Standards for reporting qualitative research (SRQR) checklist. This study reported in accordance with the SRQR checklist. We have confirmed that it is complete and added it as a supplementary file.

Add ethics information.

Author’s re

---

## [Decision Letter · Decision Letter 1]

16 Mar 2026

Towards sustainable pharmacy practice: community pharmacists’ experiences with medicine waste, reuse and disposal

PONE-D-25-44528R1

Dear Dr. Rajiah,

We’re pleased to inform you that your manuscript has been judged scientifically suitable for publication and will be formally accepted for publication once it meets all outstanding technical requirements.

Kind regards,

Muhammad Shahzad Aslam, Ph.D.,M.Phil., Pharm-D

Academic Editor

PLOS One

Additional Editor Comments (optional):

Reviewers' comments:

Reviewer's Responses to Questions

**Comments to the Author**

1. If the authors have adequately addressed your comments raised in a previous round of review and you feel that this manuscript is now acceptable for publication, you may indicate that here to bypass the “Comments to the Author” section, enter your conflict of interest statement in the “Confidential to Editor” section, and submit your "Accept" recommendation.

Reviewer #2: All comments have been addressed

2. Is the manuscript technically sound, and do the data support the conclusions?

Reviewer #2: Yes

3. Has the statistical analysis been performed appropriately and rigorously? 

Reviewer #2: Yes

4. Have the authors made all data underlying the findings in their manuscript fully available?

Reviewer #2: Yes

5. Is the manuscript presented in an intelligible fashion and written in standard English?

Reviewer #2: Yes

6. Review Comments to the Author

Reviewer #2: Thank you to the authors for their detailed and thoughtful revisions. The paper is stronger and clearer in its message and has an important place in the literature. I have no further comments.

7. PLOS authors have the option to publish the peer review history of their article (what does this mean?). If published, this will include your full peer review and any attached files.

Reviewer #2: **Yes:**Kaitlyn E Watson

---

## [Editor Report · Acceptance letter]

PONE-D-25-44528R1

PLOS One

Dear Dr. Rajiah,

I'm pleased to inform you that your manuscript has been deemed suitable for publication in PLOS One. Congratulations! Your manuscript is now being handed over to our production team.

Kind regards,

on behalf of

Dr. Muhammad Shahzad Aslam

Academic Editor

PLOS One